# Activation of a Ductal-to-Endocrine Transdifferentiation Transcriptional Program in the Pancreatic Cancer Cell Line PANC-1 Is Controlled by RAC1 and RAC1b through Antagonistic Regulation of Stemness Factors

**DOI:** 10.3390/cancers13215541

**Published:** 2021-11-04

**Authors:** Paula Marie Schmidtlein, Clara Volz, Alexander Hackel, Isabel Thürling, Darko Castven, Rüdiger Braun, Ulrich Friedrich Wellner, Björn Konukiewitz, Gabriela Riemekasten, Hendrik Lehnert, Jens-Uwe Marquardt, Hendrik Ungefroren

**Affiliations:** 1First Department of Medicine, University Hospital Schleswig-Holstein, Campus Lübeck, D-23538 Lübeck, Germany; paula.schmidtlein@student.uni-luebeck.de (P.M.S.); clara.volz@student.uni-luebeck.de (C.V.); isabel.thuerling@student.uni-luebeck.de (I.T.); castvendarko@gmail.com (D.C.); Jens.Marquardt@uksh.de (J.-U.M.); 2Department of Rheumatology and Clinical Immunology, University Hospital Schleswig-Holstein, Campus Lübeck, D-23538 Lübeck, Germany; AlexanderMaximilian.Hackel@uksh.de (A.H.); Gabriela.Riemekasten@uksh.de (G.R.); 3Clinic for Surgery, University Hospital Schleswig-Holstein, Campus Lübeck, D-23538 Lübeck, Germany; ruediger.braun@uksh.de (R.B.); ulrich.wellner@uksh.de (U.F.W.); 4Institute of Pathology, University Hospital Schleswig-Holstein, Campus Kiel, D-24105 Kiel, Germany; Bjoern.Konukiewitz@uksh.de; 5University of Salzburg, A-5020 Salzburg, Austria; hendrik.lehnert@plus.ac.at; 6Center of Brain, Behavior and Metabolism (CBBM), University of Lübeck, D-23538 Lübeck, Germany

**Keywords:** pancreatic ductal adenocarcinoma, PANC-1, quasimesenchymal, ductal-to-endocrine transdifferentiation, pancreatic β cell, stemness, pluripotency, RAC1, RAC1b, SOX2

## Abstract

**Simple Summary:**

For patients with metastatic pancreatic ductal adenocarcinoma (PDAC) there is currently no cure; hence, novel effective therapies are desperately needed. Among PDAC patients, the tumor cell phenotypes are heterogeneous as a result of epithelial–mesenchymal transition, a process that endows them with the ability to metastasize, resist therapy, and generate cancer stem cells. The heightened plasticity of quasimesenchymal and potentially metastatic tumor cells may, however, also be exploited for their transdifferentiation into benign, highly differentiated or post-mitotic cells. Since PDAC patients often have a need for replacement of insulin-producing cells, conversion of tumor cells with a ductal/exocrine origin to endocrine β cell-like cells is an attractive therapeutic option. Successful transdifferentiation into insulin-producing cells has been reported for the quasimesenchymal cell line PANC-1; however, the mechanistic basis of this transformation process is unknown. Here, we show that the small GTPases, RAC1 and RAC1b control this process by antagonistic regulation of stemness genes.

**Abstract:**

Epithelial–mesenchymal transition (EMT) is a driving force for tumor growth, metastatic spread, therapy resistance, and the generation of cancer stem cells (CSCs). However, the regained stem cell character may also be exploited for therapeutic conversion of aggressive tumor cells to benign, highly differentiated cells. The PDAC-derived quasimesenchymal-type cell lines PANC-1 and MIA PaCa-2 have been successfully transdifferentiated to endocrine precursors or insulin-producing cells; however, the underlying mechanism of this increased plasticity remains elusive. Given its crucial role in normal pancreatic endocrine development and tumor progression, both of which involve EMT, we analyzed here the role of the small GTPase RAC1. Ectopic expression in PANC-1 cells of dominant negative or constitutively active mutants of RAC1 activation blocked or enhanced, respectively, the cytokine-induced activation of a ductal-to-endocrine transdifferentiation transcriptional program (deTDtP) as revealed by induction of the NEUROG3, INS, SLC2A2, and MAFA genes. Conversely, ectopic expression of RAC1b, a RAC1 splice isoform and functional antagonist of RAC1-driven EMT, decreased the deTDtP, while genetic knockout of RAC1b dramatically increased it. We further show that inhibition of RAC1 activation attenuated pluripotency marker expression and self-renewal ability, while depletion of RAC1b dramatically enhanced stemness features and clonogenic potential. Finally, rescue experiments involving pharmacological or RNA interference-mediated inhibition of RAC1 or RAC1b, respectively, confirmed that both RAC1 isoforms control the deTDtP in an opposite manner. We conclude that RAC1 and RAC1b antagonistically control growth factor-induced activation of an endocrine transcriptional program and the generation of CSCs in quasimesenchymal PDAC cells. Our results have clinical implications for PDAC patients, who in addition to eradication of tumor cells have a need for replacement of insulin-producing cells.

## 1. Introduction

Pancreatic ductal adenocarcinoma (PDAC) is a highly aggressive cancer with an extremely poor prognosis [1,2]. This is mainly due to locally advanced or metastatic disease that the majority of patients present with at the time of diagnosis and which reduces the number of patients amenable to surgery—the only curative option currently—to about one-fifth. Human PDAC, which is thought to originate from the exocrine cells of the pancreatic ducts, is a heterogeneous disease, which is classified into subtypes that differ with respect to functional behavior in preclinical models, survival in clinical studies, as well as response to drug treatment. The most malignant PDAC molecular subtype is referred to as quasimesenchymal (QM) or basal-like/squamous [3]. The cancer cells of this subtype have acquired a mesenchymal phenotype during the process of epithelial–mesenchymal transition (EMT). EMT is driven by master transcription factors, i.e., SNAIL1 and SNAIL2/SLUG, encoded by the SNAI1 and SNAI2 genes, respectively, to result in reduced membrane expression of the epithelial marker E-cadherin and upregulation of mesenchymal markers such as vimentin. In many epithelial tissues the EMT program is associated with the generation of cancer stem cells (CSCs) [4], which in PDAC promote tumor growth, metastatic spread, therapy resistance, or tumor relapse post-therapy [5].

An increasing number of studies in experimental animals and humans suggests that normal (or neoplastic) pancreatic ductal or acinar cells can be converted to insulin-producing cells, i.e., in response to stimuli such as pancreatic injury, inflammation, metabolic stress, or β cell loss [6]. Pancreatic cell interconversions depend on Neurogenin 3 (Ngn3), a key endocrine progenitor transcription factor necessary for the specification of endocrine cells [7]. During endocrine progenitor maturation Ngn3-expressing cells co-express E-cadherin and vimentin [8], while Ngn3 represses E-cadherin through induction of Slug [9,10], together suggesting that early pancreas development involves EMT-like programs. Due to the intimate connection between the EMT process and CSCs, tumor cells with a mesenchymal phenotype are considered to be more plastic as inferred from their higher potential for transdifferentiation to other cell lineages [4,11,12]. For instance, the pancreatic ductal cell line, PANC-1, derived from a QM-type adenocarcinoma, is the only pancreatic cancer cell line for which transdifferentiation into endocrine precursors with NGN3 activation [6,13] or β cell-like cells [14,15,16,17,18,19,20,21,22] has been reported. The different protocols for generation of insulin-producing cells involved the use of either growth factors [14,15,16], proteinase-activated receptor agonists [17], thyroid hormones [18], the diterpene lactone andrographolide [19], transfection with an *INS* promoter or Insulinoma-associated antigen-1 zinc-finger transcription factor [20], high-glucose [21] or RGD-covered dextran derivative surfaces [22]. The variety of agents used to achieve this along with the lack of equivalent studies in other cell lines suggests that PANC-1 cells are particularly responsive to differentiation cues. We have recently compared a panel of E and QM PDAC cell lines for their capacity to activate a ductal-to-endocrine transdifferentiation transcriptional program (deTDtP) using a combination of fibroblast growth factor-basic (FGFb) and transferrin (TF) as inducers [14]. While E type cell lines were not or only weakly responsive with regard to induction of the genes for Insulin (encoded by *INS*), Glucose transporter-2 (GLUT-2, encoded by *SLC2A2*), and MafA (encoded by *MAFA*), the QM lines PANC-1 and MIA PaCa-2 reacted strongly [23]. However, the molecular basis of this high deTD potential of PANC-1 (and MIA PaCa-2) cells, i.e., with regard to the involvement of stem cell markers or their upstream regulators, remains obscure. 

An important driver of physiological and pathophysiological EMT programs in the pancreas is the small GTPase, Rac1 [24,25]. In pancreatic islet cell differentiation, Rac1 controls delamination and migration through modulating E-cadherin-mediated cell–cell adhesion [8], and in the mature β cell Rac1 signaling is involved in regulating the second-phase of insulin secretion [26]. In solid tumors, RAC1 participates in EMT phenotype switching, CSC generation, and in PDAC initiation through reprogramming of mutant KRAS-expressing acinar cells that spontaneously undergo acinar-to-ductal metaplasia (ADM) [27,28]. These findings suggest that RAC1 may be a potential target in suppressing EMT-associated plasticity involved in both normal pancreatic endocrine development and malignant progression of PDAC.

Besides RAC1 tumor cells from pancreatic, breast, colon, lung, and thyroid cancers express an alternatively spliced isoform of the human RAC1 gene, designated RAC1b, that differs from RAC1 by the inclusion of an additional exon (reviewed in [29]). In established cell lines of PDAC and breast cancer, RAC1b protein expression was strongly associated with a well-differentiated phenotype/E state in contrast to the parental RAC1 isoform [30,31]. Consistent with higher expression in E cells, RAC1b promoted the expression of CDH1, CLDN7, and KRT19, and blocked that of VIM [30]. In PANC-1 cells, RAC1b potently inhibited TGF-β1-dependent EMT [32] and the associated upregulation of *SNAI1* and *SNAI2* [30,32], the latter of which is involved in both pancreatic endocrine specification [10] and CSC maintenance, self-renewal, and function [33,34,35]. Hence, it is conceivable that through its potent anti-EMT function RAC1b is capable of inhibiting CSCs and cellular plasticity of ductal epithelial cells.

Given the involvement of RAC1 in EMT-associated plasticity in both normal pancreatic endocrine development and malignant progression of PDAC, we studied here in more detail its role in deTD. Moreover, the ability of RAC1b to protect PDAC-derived cells from TGF-β1-induced mesenchymal conversion [30,32] along with its low or absent expression in PANC-1 and other QM PDAC-derived cells [30] suggested the possibility that RAC1b acts an endogenous inhibitor of RAC1-dependent endocrine deTD in PANC-1 cells. We also studied whether RAC1 and RAC1b impact on stem/pluripotency genes to control cellular plasticity. 

## 2. Results

### 2.1. Inhibition of RAC1 Activation Interferes with a deTDtP into Pancreatic Endocrine Progenitors and Insulin-Expressing Cells

RAC1 has been implicated in islet cell delamination and cell migration, promotion of EMT, and CSC generation. To reveal whether this small GTPase is causally involved in driving plasticity and deTD in QM-type PDAC cells, we employed previously characterized PANC-1 cells stably expressing RAC1^N17^ [36], an inactive mutant that acts in a dominant-negative (dn) fashion to block the action of endogenous wild-type RAC1. The expression and functional activity of the N17 mutant were verified here by immunoprecipitation (Appendix A) and its ability to reduce basal migration of PANC-1 cells (Appendix A) mimicking the loss in migratory activity after siRNA-mediated RAC1 knockdown [36]. Following application of an FGFb + TF-based protocol [14] for induction of a β cell transcriptional program—assessed by expression of INS, SLC2A2, and MAFA [23]—we noted that the induction of these markers was clearly impaired upon dn inhibition of RAC1 activation (Figure 1A). Similar results for *INS* expression were obtained with MIA PaCa-2 cells transiently transfected with an RAC1^N17^-encoding expression vector (Figure 1B and Appendix A).

To confirm the inhibitory function of RAC1^N17^ we employed an alternative protocol for deTD that used a 3-day treatment with a combination of inflammatory cytokines IFNγ, IL1β, and TNFα (IIT) [6,23]. This cocktail was originally shown to induce expression of the endocrine progenitor marker NGN3 (encoded by *NEUROG3*) in (parental) PANC-1 cells [6]. However, we observed that it was also able to transcriptionally induce β cell-specific markers, i.e., insulin, the mRNA levels of which remained stable for up to 7 days post-treatment (Appendix A). Moreover, an analysis of seven single cell-derived clones of PANC-1 cells subjected to the same deTD protocol revealed that *INS* was induced in all clones albeit to a different extent (Appendix A). When applied to PANC-1^RAC1-N17^ these cells responded much weaker than empty vector controls with respect to activation of *NEUROG3* (Figure 1C) as well as *INS*, *SLC2A2,* and *MAFA* (Figure 1D).

Next, we asked whether transiently transfecting PANC-1 cells with a constitutively active RAC1 mutant (L61, expression verified in Appendix A, right-hand panel) would increase the cells’ propensity to convert into insulin-expressing cells upon deTD culture (TDC) with either FGFb + TF or IIT. Intriguingly, under both transdifferentiation conditions ectopic expression of RAC1^L61^ resulted in elevated levels of mRNA transcripts for *INS* (Figure 1E) and *SLC2A2* relative to empty vector transfected control cells. These data show for the first time that RAC1 is crucial for activation of deTDtP transformation in QM-type PDAC-derived cells.

### 2.2. Ectopic RAC1b Suppresses Transdifferentiation of PANC-1 Cells into Pancreatic Endocrine Progenitors and Insulin-Expressing Cells

We have previously shown that the RAC1 splice isoform, RAC1b, acts as an endogenous inhibitor of RAC1 in the regulation of EMT and cell migration in PDAC-derived cells, consistent with low or absent expression of RAC1b in the QM cell lines PANC-1, MIA PaCa-2, and PaTu 8988s [23,30,32]. To study if RAC1b also impacts the deTDtP, we ectopically expressed in a stable or transient fashion an HA-tagged version of RAC1b in PANC-1 or MIA PaCa-2 cells, respectively. PANC-1 cells stably expressing RAC1b (PANC-1^HA-RAC1b^) have been characterized previously and shown to exhibit reduced migratory activity [37]. When PANC-1^HA-RAC1b^ and empty vector control cells were exposed to 5′-aza-2′-deoxycytidine (5′-Aza) for 3 d—a treatment that induces NGN3 expression [13]—the transfected RAC1b gene prevented upregulation of *NEUROG3* (Figure 2A). 

Next, the same cells were subjected to β cell-directed TDC with FGFb + TF as described above, and expression of *INS* was assessed by qPCR. Of note, expression of insulin in HA-RAC1b transfectants was decreased after TDC relative to empty vector controls (Figure 2B). Neither 5′-Aza nor FGFb + TF treatment altered the protein expression level of HA-RAC1b (Appendix A). Together, these data suggest that, in contrast to RAC1, RAC1b impairs the deTD potential of QM cells to pancreatic endocrine progenitors and insulin-expressing cells.

### 2.3. RAC1b Knockout Enhances the deTDtP to Pancreatic Endocrine or Insulin-Expressing Cells

Next, we generated PANC-1 cells that lack RAC1b protein by removing the RAC1b-specifying exon 3b from RAC1 using CRISPR/Cas9 technology in combination with retroviral infection to generate PANC-1^RAC1b-KO^ cells. These cells have been characterized previously and shown to lack RAC1b protein but retain unaltered RAC1 expression [38]. To study the impact of RAC1b depletion on the susceptibility of PANC-1 cells to deTD, PANC-1^RAC1b-KO^ and control cells were treated with 5′-Aza and evaluated for expression of *NEUROG3* by qPCR. Of note, PANC-1^RAC1b-KO^ cells responded much more vigorously with activation of this gene than controls (Figure 3A). 

To analyze if RAC1b also impacts TD to the β cell phenotype, we subjected PANC-1^RAC1b-KO^ cells to FGFb + TF-based TDC, as performed in Figure 2, and determined transcriptional activities of *INS*, *SLC2A2,* and *MAFA*. As shown in Figure 3B, expression of these genes was clearly enhanced in RAC1b-KO cells compared to control cells. Together with the data presented in Figure 2, we conclude that RAC1b inhibits transcriptional changes that convert PANC-1 cells into pancreatic endocrine precursors and β cell-like cells.

### 2.4. Inhibition of RAC1 Activation Decreases Pluripotency Marker Expression and the Generation of CSC-like Cells

Given the requirement of RAC1 activation for deTD transcriptional changes in PDAC cells and the functional relationship between activated RAC1 and CSCs, we asked whether inhibition of RAC1 activation impacts pluripotency gene expression in PANC-1 cells. To this end, qPCR-based measurement revealed that ectopic RAC1^N17^ expression decreased the abundance of NANOG and SOX2 mRNAs in PANC-1 cells compared to empty vector controls (Figure 4A). While NANOG protein levels in PANC-1 and PANC-1^RAC1-N17^ cells were below the detection limit in immunoblots, SOX2 protein was readily detectable, and its abundance was reduced in RAC1^N17^-expressing cells (Figure 4B and Appendix A). Conversely, transient transfection of constitutively-active RAC1^L61^ resulted in upregulation of SOX2 protein levels (Figure 4C and Appendix A). Dn RAC1 also decreased expression of CD326/EpCAM and CD133, two established CSC markers in PDAC [39], as determined by flow cytometry analysis (Figure 4D).

Next, we employed the colony formation assay (CFA) to assess the number of cells with stem cell features by virtue of their ability to generate from a single cell a microscopically discernible clone or colony [40]. A quantitative analysis by counting revealed that PANC-1^RAC1-N17^ cells generated fewer colony forming units (CFUs) of lower size than vector controls did (Figure 4E). We conclude that RAC1 in its activated form is crucial for promoting pluripotency gene expression and clonogenic potential in PANC-1 cells.

### 2.5. Selective Knockout of RAC1b Enhances Pluripotency Marker Expression and the Generation of CSC-like Cells

We have previously shown that RAC1b is a powerful inhibitor of EMT and cell motility in PDAC cells and thus acts as a functional antagonist of RAC1. Given the close connections among RAC1, CSCs, and EMT, we pursued the idea that the inhibitory function of RAC1b extends to transcriptional control of stem cell genes and self-renewal. Monitoring the expression of various pluripotency and stem cell markers by qPCR revealed upregulation in PANC1^RAC1b-KO^ cells of *POU5F1* (encoding OCT4, specifically the stem cell-associated OCT4A isoform) [42], *NANOG*, *SOX2*, *KLF4*, and *ZFP42* (Figure 5A). The ZFP42/REX1 protein inhibits embryonic stem cell differentiation, and its expression is regulated by Sox2, Nanog, and Oct4 [43]. At the protein level, we observed increased abundance in PANC-1^RAC1b-KO^ cells of OCT4A (Figure 5B and Appendix A) and of NANOG and SOX2, the levels of which decreased upon TDC, consistent with cellular differentiation to β cell-like cells (Figure 5B and Appendix A). When single cell-derived clones of PANC-1^RAC1b-KO^ cultures were immunoblotted for SOX2, it was observed that all clones presented with increased SOX2 and OCT4A protein expression albeit with strong variations among clones (Appendix A). RAC1b-deficient PANC-1 cells also expressed more CD326 as determined by flow cytometry (mean ± SD, PANC-1^RAC1b-KO^: 75.1% ± 8.3% vs. Control: mean ± SD, 50.6% ± 4.3%, *p* = 0.0297) (Appendix A). To reveal if PANC-1^RAC1b-KO^ cells exhibit increased clonogenic potential, we performed CFAs and, in addition, spheroid formation assays (SFAs) to measure the number of stem-like cells based on their ability to grow and form colonies (spheroids) in suspension. The loss of RAC1b in PANC-1 cells resulted in a dramatic increase in both the number of CFUs in CFAs (Figure 5C) and of spheroid forming units (SFUs) in SFAs (Figure 5D). In addition, the average size of the CFUs was increased (Figure 5C) and the spheres were larger and more compact (Figure 5D).

CSCs may serve as seeds for tumor metastasis at secondary sites. Intriguingly, upon depletion of RAC1b, we observed in PANC-1 cells re-expression of RUNX3 (Appendix A), a transcription factor that has been shown in PDAC to orchestrate a metastatic program involving stimulation of cell invasion and secretion of proteins that favor distant colonization [44]. We conclude that RAC1b potently inhibits pluripotency gene expression and the generation/self-renewal of CSC-like cells.

### 2.6. Pharmacological Inhibition of RAC1 Activation Relieves the RAC1b-Knockout-Induced Transdifferentiation into Insulin-Expressing Cells

Prompted by the data above, we hypothesized that RAC1 and RAC1b antagonistically regulate the deTD potential in PANC-1 cells. To more directly evaluate this possibility, we performed mutual rescue experiments with established RAC1 and RAC1b inhibitors. First, we asked whether selective knockdown of RAC1b impacts the suppressing effect of RAC1^N17^ on the deTDtP. To this end, transfection of PANC-1 cells with a RAC1b/exon 3b-selective small interfering RNA (siRNA), but not an irrelevant control siRNA, strongly increased induction of the INS, SCLC2A2, and MAFA genes in PANC-1^RAC1-N17^ cells (Figure 6A). Finally, we addressed the question if the reciprocal rescue experiment, i.e., inhibition of RAC1 in PANC-1^RAC1b-KO^ cells (which present with strongly elevated induction of these genes, see Figure 3), would protect these cells from this induction. Since selective knockdown of RAC1 by RNA interference was not feasible, given that all nucleic acid sequences of RAC1 are also present in RAC1b, we employed instead NSC 23766, an established inhibitor of RAC1 activation [45,46]. In accordance with our assumption, the induction of the deTDtP in PANC-1^RAC1b-KO^ cells was partially blocked in the presence of NSC 23766, as evidenced by a failure of these cells to upregulate *INS*, *SLC2A2,* and *MAFA* to the same extent as control cells, which received transdifferentiation medium without NSC 23766 (Figure 6B). We interpret these findings to indicate that RAC1b depletion has removed a barrier to the pro-stem cell effect of RAC1 from which the cells can be rescued by inhibition of RAC1 activation. Hence, the potent suppression of CSC function and pluripotency by RAC1b counteracts the respective promoting effects of RAC1, and the activities of both isoforms balance each other to regulate transcriptional plasticity and the propensity for deTD.

## 3. Discussion

Based on the idea that exocrine ductal and endocrine β cells share a common stem cell, the PDAC-derived cell line, PANC-1, was successfully converted to insulin-producing/expressing cells using a variety of protocols [14,15,16,17,18,19,20,21,22,23]. To reveal whether this ability was confined to only this cell line, ductal cells with a QM phenotype, or was a general feature of pancreatic ductal cells regardless of their subtype, we compared in a separate study the response of various PDAC cell lines of either E or QM phenotype to pancreatic endocrine- or pancreatic β cell-specific transdifferentiation using 5′-Aza and FGFb + TF-based protocols. We observed that another QM PDAC cell line besides PANC-1, MIA PaCa-2, possesses a much higher deTD potential than all E type lines tested; however, PANC-1 cells stood out for being the most responsive [23]. Of note, although considered to be QM, PANC-1 cells in contrast to MIA PaCa-2 cells retained some epithelial characteristics, i.e., low but readily detectable levels of E-cadherin and RAC1b expression [30,32], consistent with the current view that cells with mixed expression of epithelial and mesenchymal markers and a partial EMT phenotype are the most plastic ones [4,47]. 

Given its crucial role in pancreatic islet cell differentiation [8], tumor-associated EMT phenotype switching, CSC generation, and ADM-based reprogramming of acinar cells in PDAC initiation [27,28], we speculated that RAC1 may be a good candidate for governing the endocrine transformation process and the underlying transcriptional program. To this end, we observed that dn interference with RAC1 activation in PANC-1 cells strongly decreased the FGFb + TF-induced transcriptional activation of *INS*, *SLC2A2,* and *MAFA*, suggesting reduced potential for deTD, while transient expression of constitutively active RAC1 amplified the deTD response. Earlier, we demonstrated that the RAC1 splice isoform, RAC1b, can protect cells from mesenchymal transition by TGFβ1 and an increase in TGFβ1 and RAC1-driven cell motility in both PDAC and breast cancer cells [31,32]. This suggested the possibility that RAC1b also antagonizes the RAC1-driven deTDtP in pancreatic cells. To this end, enforced ectopic expression of RAC1b in PANC-1 or MIA PaCa-2 cells strongly attenuated 5′-Aza-induced NGN3 and FGFb + TF-induced β cell marker expression. Conversely, genetic knockout of exon 3b of *RAC1* and, hence, selective depletion of RAC1b protein from PANC-1 cells dramatically increased the transcriptional activity of *NEUROG3*, *INS*, *SLC2A2*, and *MAFA*. These data support the concept of RAC1b acting as a negative regulator and natural antagonist of RAC1 in pancreatic ductal-to-endocrine conversion and acquisition of β cell identity.

A functional relationship between RAC1 and CSCs was suggested from findings in non-small cell lung adenocarcinoma (NSCLAC). Side population cells isolated from this tumor type (harboring putative CSCs) contained elevated levels of RAC1-GTP, which drove the dynamic conversion of non-CS/progenitor cells to CS/progenitor cells [46,48]. RAC1 is also involved in the maintenance of cancer stemness in glioma stem-like cells [49] and in kidney cancer in response to upregulation by a novel oncogenic long noncoding RNA, LncRNA NR2F2-AS1 [50]. In gastric adenocarcinoma, RAC1 overexpression promoted the CSC phenotype, chemotherapy resistance, and expression of SOX2 in spheroids of gastric cancer cells [45]. Consistent with this, we observed in PANC-1 cells with stable expression of RAC1^N17^ downregulation of SOX2 at both the mRNA and protein level as well as a decrease in membrane-associated expression of the pancreatic stem cell markers CD133 and CD326/EpCAM. Moreover, inhibition of RAC1 activation/GTP binding interfered with the colony-forming ability and, hence, the self-renewal capacity of PANC-1 cells. Given its powerful function as an RAC1 antagonist, we consequently extended our study to reveal whether inhibition of the deTDtP by RAC1b may be a consequence of the repression of stemness traits. Intriguingly, cellular depletion of RAC1b resulted in a dramatic rise in the number of colony- and spheroid-forming units. Consistent with this, we observed de-repression of various pluripotency genes such as *POU5F1*, *SOX2*, and *NANOG* at both the mRNA and protein level in PANC-1^RAC1b-KO^ compared to control cells. SOX2 appears to be of special importance for the transdifferentiation to insulin-producing cells, as we have previously shown that ectopic SOX2 transfection enhanced the transdifferentiation of peripheral blood monocytes into insulin-producing cells via an increase in the plasticity of the intermediate stem-like PCMO cell [42,51,52]. Sequential EMT and mesenchymal–epithelial transition (MET) steps mediating epithelial plasticity have been shown to be essential in Oct4/Klf4/Sox2/Myc (OKSM)-mediated reprogramming of murine embryonic fibroblasts into induced pluripotent stem cells (iPSCs) [53]. Interestingly, ectopic overexpression of OCT4 and NANOG in lung adenocarcinoma cells activated SLUG and enhanced sphere formation, drug resistance, tumor-initiating capability, and promoted EMT, confirming that OCT4/NANOG signaling controls EMT [54]. Slug is also expressed in Ngn3-positive endocrine progenitor cells of the developing mouse endocrine pancreas, and during endocrine cell differentiation it becomes increasingly restricted to β cells [10], while the related SNAIL contributes to the maintenance of a stem cell-like phenotype in human pancreatic cancer [34]. In agreement with induction of *POU5F1* and *NANOG* in PANC-1^RAC1b-KO^ cells, we previously observed upregulation of *SNAI1* and *SNAI2* in PANC-1 cells with either knockout or knockdown of RAC1b [32]. Here, we extended the functional connection between master regulators of EMT and CSCs [4] to RAC1 and RAC1b in the control of the deTDtP. Successful conversion of PANC-1 cells to highly differentiated cells with activation of a β cell-like transcriptional program upon FGFb + TF-based TDC is also evident from downregulation of the NANOG and SOX2 proteins (Figure 5B and Appendix A). The opposing roles of RAC1 and RAC1b on pluripotency factors and the deTDtP have been summarized in Figure 7.

The functional antagonism between RAC1 and RAC1b in the regulation of EMT and stemness/pluripotency gene expression may extend to pancreatic islet morphogenesis and islet cell migration since endocrine delamination of pancreatic endocrine cells from the developing epithelium is governed by a mixed EMT and Rac1 signaling [8]. In line with this assumption, we observed that selectively depleting RAC1b partially rescued PANC-1 cells from the RAC1^N17^-mediated decrease in FGFb + TF-induced *INS*, *SLC2A2,* and *MAFA* expression, while inhibiting RAC1 activation with the drug NSC 23766 partially rescued PANC-1cells from the RAC1b-KO-mediated increase in FGFb + TF-induced expression of these genes. It will be interesting to see which signaling pathways mediate the antagonistic effects of RAC1 and RAC1b on pluripotency genes and the deTDtP. In lung cancer cell progression, both isoforms also exhibit different signaling and functional activities. While both RAC1 and RAC1b enhanced phosphorylation of p38α, AKT, GSK3β, and activated serum response and Smad-dependent gene promoters, RAC1b only activated JNK2 and, in addition, TCF/LEF1 and NFκB-responsive genes [55]. Of note, AKT, GSK3β, Wnt/β-catenin, and TGFβ/Smad signaling have all been shown to mediate various stem cell properties, such as self-renewal, cell fate decisions, survival, proliferation, and differentiation [56].

Why does the PANC-1 cell line stand out for its high propensity for deTD? Re-activation of stemness-related pathways and alternative splicing [57] appear as significant causes of cellular plasticity because they promote the acquisition of stem-like properties through a profound phenotypic reprogramming of cancer cells. Consistent with this, we observed significant intercellular heterogeneity in PANC-1 cells with respect to basal levels of SOX2 and OCT4A proteins and TDC-induced *INS* expression. The RAC1 splice product RAC1b, which appears to prevent the RAC1-driven progression to a QM and, hence, a more plastic phenotype, is only expressed at a low level in PANC-1 cells. In addition, PANC-1 cells are known for their high autocrine production of TGFβ1 [58], which might contribute to CSC formation and plasticity. In fact, TGFβ-induced EMT increases cancer stem-like cells in PANC-1 cells [59], which fits observations that side population cells could be isolated from monolayers of PANC-1 but not E type BxPC-3 cells [60] and predominate in TGFβ1-induced EMT and invasion [61]. A mechanistic association between TGFβ1, TNFα, and the CSC properties has also been observed in MIA PaCa-2 cells [62]. Finally, under hyperglycemic conditions, chronic exposure of pancreatic epithelial cells can induce autocrine TGFβ production with subsequent induction of EMT and stem cell traits such as activation of *NANOG* [63]. The non-canonical arm of TGFβ signal transduction in PDAC is known to involve RAC1 signaling, i.e., RAC1 was discussed as a potential mediator of TGFβ-stimulated invasion in PDAC organoids with mutant DPC4 [64], and in lung adenocarcinoma A549 cells a NSC 23766-induced decrease in RAC1 activity partially blocked TGFβ1-induced EMT and the increase in ALDH^high^ CSCs [48]. Since RAC1 GTPase activity is critical for non-CS/progenitor cell to CS/progenitor cell transition in NSCLAC (see above), increasing the cellular levels of RAC1b, and hence the ratio of RAC1b:RAC1, might represent an innovative therapeutic strategy to reduce CSC plasticity [46]. Intriguingly, stimulation of PANC-1 cells with exogenous TGFβ1 downregulated RAC1b but not RAC1 protein expression (H.U., unpublished observation), suggesting that the induction of pluripotency genes and phenotypic plasticity is a (direct) consequence of TGFβ’s ability to lower the RAC1b:RAC1 ratio. Other signals that may impact this ratio may come from the tumor microenvironment in the form of proinflammatory agents and vascular endothelial growth factor [65]. In glioblastoma and hepatocellular carcinoma, a hypoxic microenvironment maintains stem cells or promotes reprogramming towards a CSC phenotype, i.e., by inducing EMT via activation of *SNAI1* through HIF-1α [65].

EMT most frequently occurs at the invasive front of the primary tumor in anatomical structures called tumor buds [66]. As speculated earlier [23], a therapeutically intended deTD approach may therefore be particularly effective in preventing the very early stages of tumor cell dissemination of potentially metastatic PDAC cells. In case the resulting β cell-like cells are capable of glucose-regulated insulin secretion, this transdifferentiation strategy might be applicable to PDAC patients with a long-term history of type 2 diabetes and insulin deficiency. Moreover, epigenetic reprogramming of PANC-1 cells with 5′-Aza did not only increase NGN3, INS, and SST expression but also induced a less aggressive phenotype with impaired tumor growth and improved response to the cytotoxic drug gemcitabine [67]. Our results suggest that for the treatment of PDAC and PDAC-related diabetes, therapeutically modulating the RAC1b:RAC1 ratio may be feasible in both instances, prevention of dedifferentiation/eradication of CSCs (through an increase) or enhancing dedifferentiation and stem cell features (through a decrease) for subsequent deTD therapy.

## 4. Material and Methods

### 4.1. Generation of PANC-1^RAC1-N17^, PANC-1^RAC1-L61^, PANC-1^HA-RAC1b^, PANC-1^RAC1b-KO^, and PANC-1^RAC1b-KD^ Cells

The PDAC-derived cell lines PANC-1 and MIA PaCa-2 were maintained in standard growth medium consisting of RPMI 1640, 10% FBS, 1% glutamine, 1% sodium pyruvate, and penicillin/streptomycin [30,32]. The generation of PANC-1 cells ectopically expressing either MYC-RAC1^N17^ (in pRK5) [36] or HA-RAC1b (in pCGN) [37], or PANC-1 cells devoid of exon 3b of RAC1 and hence the ability to produce RAC1b protein [38], was described in detail earlier. Ectopic expression of the transgene or lack of RAC1b protein, respectively, as well as functionality of the resulting mutants has been validated in several studies from our group [37,38].

To generate PANC-1^RAC1-L61^ or PANC-1 cells with selective RAC1b knockdown (PANC-1^RAC1b-KD^), PANC-1 cells were transiently transfected with either MYC-RAC1^L61^ (in pRK5) or a RAC1b-specific siRNA using Lipofectamine 2000 (Invitrogen/Thermo Fisher Scientific, Dreieich, Germany) as described in detail earlier [38]. The functionality of the RAC1b siRNA was demonstrated in previous publications [37,38].

### 4.2. Quantitative Real-Time Reverse Transcription-PCR

The procedure and the conditions for quantitative real-time Reverse Transcriptase (RT)-PCR (qPCR), which was performed on an I-cycler with IQ software (Bio-Rad, Munich, Germany), were published earlier [30,32,38,58]. Primers were generally chosen to span exon–intron boundaries. For amplification of the NANOG, POU5F1, and SOX2 genes, which lack introns, RNAs were treated with DNase I (Invitrogen/Thermo Fisher Scientific) prior to reverse transcription. All values for the genes of interest were normalized to those for glycerinaldehyde-3-phosphate-dehydrogenase (GAPDH) and TATA box-binding protein (TBP), and relative gene expression was calculated by the 2-∆∆Ct method. Sequence information for amplification primers for the pluripotency genes [42] and the pancreatic endocrine and β cell-specific genes [51,68] was provided earlier and is summarized in Appendix A.

### 4.3. Immunoblotting

Cells were lysed in PhosphoSafe buffer (Calbiochem/Merck Millipore, Taufkirchen, Germany), and the protein concentrations were determined with the Bradford assay (PIERCE, Rockford, IL, USA). Equal amounts of cellular proteins were fractionated by SDS-PAGE, transferred to PVDF membrane, and immunoblotted as described in detail earlier [37,38]. Except for anti-HSP90, #13119, Santa Cruz (Heidelberg, Germany) and anti-Rac1, #610650, BD Transduction Laboratories (Heidelberg, Germany), all antibodies used were from Cell Signalling Technology (Frankfurt/Main, Germany): GAPDH (14C10), #2118; NANOG (D73G4), #4903; Oct4, #2750; RUNX3/AML2 (D9K6L), #13089; and Sox2, #2748. 

### 4.4. Flow Cytometric Analysis

The membrane-associated markers CD133 and CD326/EpCAM were measured by flow cytometry with the following fluorochrome-conjugated antibodies purchased from Miltenyi Biotech (Bergisch-Gladbach, Germany): CD326/EpCAM-APC, human (clone: REA764, #130-098-118) and CD133/1 (AC-133)-PE (#130-098-826). Isotype antibodies were used as controls. Prior to antibody binding, cells were grown to confluency and maintained in medium with 0.5% FBS for at least one week to reduce the effect of serum. Following detachment with Biotase (Merck/Biochrom, Darmstadt, Germany and cell counting, 100,000 cells of PANC-1^RAC1-N17^, PANC-1^RAC1b-KO^, or the respective controls were incubated with antibodies diluted in phosphate-buffered saline (PBS)/3% human serum for 20 min. Analysis was performed on a FACSCanto II flow cytometer (BD Biosciences, Heidelberg, Germany) and Diva Software v9.0 (BD Biosciences).

### 4.5. Colony Formation and Spheroid Formation Assays

To assess their self-renewal potential, cells were subjected to CFAs, the details of which were given earlier [40,69]. Briefly, PANC-1^RAC1b-KO^ and control cells were detached with Accutase (Sigma, Deisenhofen, Germany), resuspended in culture medium, and precisely 400 viable cells/6-well in triplicate were plated for each condition. After incubation for 10 d the medium was aspirated, and colonies were washed once in PBS and fixed with 1 mL of paraformaldehyde (4.5%, *w/v*) for 15 min. Colonies were then stained with 0.1% crystal violet for 1 h, washed twice in ddH2O, air-dried overnight, and counted when consisting of at least 50 cells.

For the soft agar spheroid formation assay, 2% agarose made in PBS was autoclaved and kept in the water bath at 47ºC. Next, for the plate preparation (48-well), 6 mL of DMEM containing 20% FBS was mixed with 4 mL of 2% agarose in a 15 mL falcon tube to yield a final concentration of 1.3% agarose. A total of 300 μL of this mix was added to each well of a 48-well plate and left to harden for 5 min as the base layer (diluted to the final concentration of 0.25%). Next, experimental cells were rinsed in PBS, trypsinized, and counted using BioRad cell counter TC20 (Bio-Rad). A total of 10,000 viable cells were first suspended in 1 mL of DMEM (20% FBS), and the cell suspension was mixed with 2% agarose (700 μL containing 7000 cells + 100 μL agarose). From this suspension, 120 μL (containing 1000 cells) was added on top of the base layer in the 48-well plate (diluted to a final concentration of 0.25%). The medium was carefully replaced every 3 d. After a period of 14 d, the wells with spheroids (5 replicates in each experiment) were counted manually under a microscope.

### 4.6. Transdifferentiation of PANC-1 Cells into Pancreatic Endocrine Progenitor Cells and Insulin-Expressing Cells and Inhibitor Treatment

TD along the endocrine path was carried out according to a protocol by Lefebvre and colleagues [13]. In brief, PANC-1 mutants, vector controls, or MIA PaCa-2 cells were treated with 5′-Aza (Sigma) for 3 d in standard growth medium followed by another 3 d in medium without 5′-Aza. PANC-1 and MIA PaCa-2 cells can be induced by growth factors to form aggregates that subsequently differentiate into hormone-expressing islet-like cell aggregates [14,23]. At ~70% confluence, the growth medium was removed, and cells were washed and then treated with 50% trypsin/PBS for 1 min to loosen but not to detach the cellular monolayer from their extracellular matrix. Cells were then incubated in serum-free RPMI supplemented with 0.1% BSA, 1.1 µg/mL TF (Sigma), and 500 ng/mL recombinant human FGFb (Preprotech, Hamburg, Germany). Modifications to the original protocol [14] include the replacement of DMEM by RPMI, the omission of prestimulation with high-glucose (4.5 g/L) and addition of human placental lactogen isoform A. Preliminary experiments indicated that high-glucose did not enhance deTD, probably because this stimulates secretion of TGFβ from pancreatic cancer cells [61], which in turn has been shown to interfere with β cell differentiation [70,71]. In some assays, the FGFb + TF in serum-free medium was replaced by IFNγ, IL1β, and TNFα (all from Preprotech) and standard growth medium containing 10% FBS [6] or PANC-1^RAC1b-KO^ cells with NSC 23766 (Calbiochem). This drug is a water-soluble, cell-permeable, and reversible inhibitor of Rac1 GDP/GTP exchange, which blocks the interaction between Rac1 and the Rac-specific guanine nucleotide exchange factors, Trio and Tiam1. In some experiments, the culture time of parental PANC-1 cells treated for 3 d with IIT was extended for up to 7 d. Specifically, the IIT mix was removed, and cells were briefly rinsed in PBS followed by addition of standard growth medium (without proinflammatory cytokines) for various lengths of time.

### 4.7. Statistics 

Statistical significance was calculated either from at least three independent experiments using the Wilcoxon test or from at least three technical replicates of a representative experiment with the two-tailed unpaired Student’s *t*-test. Data were considered significant at *p* < 0.05 denoted in the graphs by an asterisk (*).

## 5. Conclusions

The ductal QM-type PDAC cell line PANC-1 is known for its high propensity to activate a pancreatic β cell-specific transcriptional program in response to various cues; however, the molecular mechanism(s) underlying the deTDtP has remained elusive. Our findings that deTD in these cells is controlled by RAC1 and RAC1b through opposing actions on pluripotency/stem cell markers provide a rationale for therapeutically shifting their balance towards RAC1 in order to facilitate endocrine conversion (Figure 7). The next step will require evaluation in mouse models of whether genetically or pharmacologically engineering the ratio of RAC1:RAC1b activity in PDAC cells impacts conversion to bona fide β cells in vivo. Since PDAC patients often have a need for replacement of insulin-producing β cells, i.e., due to tumor-induced destruction of pancreatic islets or long-standing chronic type 2 diabetes associated with hypoinsulinemia, in vivo transdifferentiation of metastatic tumor cells to insulin-producing cells represents a “two birds with one stone” strategy for fighting cancer and restoring glucose homeostasis. 

## Figures and Tables

**Figure 1 cancers-13-05541-f001:**
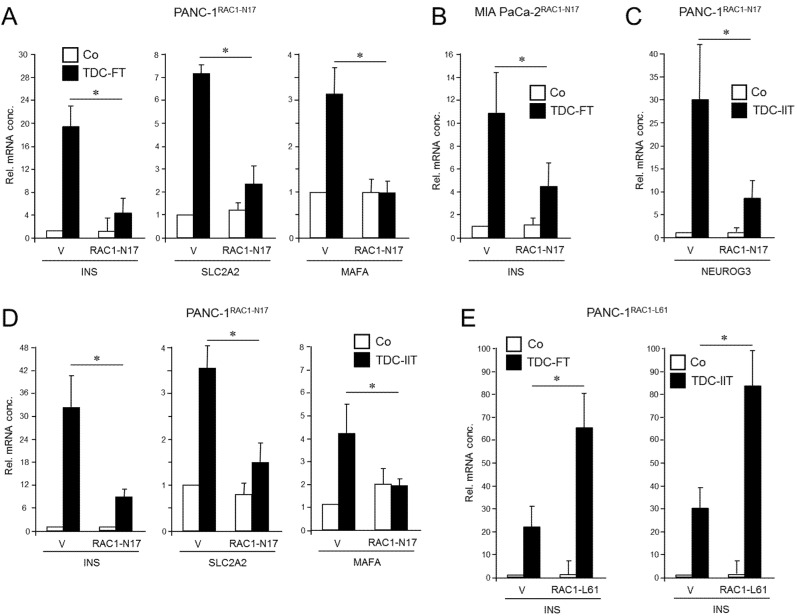
Effect of ectopic expression of non-activatable and constitutively active forms of RAC1 on a deTDtP in PANC-1 and MIA PaCa-2 cells. (**A**) PANC-1 cells stably expressing RAC1^N17^ underwent TDC with FGFb + TF (FT) for 5 d and after lysis were monitored for β cell-specific gene expression by qPCR. (**B**) MIA PaCa-2 cells were transiently transfected with RAC1^N17^ (or empty pRK5 vector) and subsequently subjected to TDC with FT for 5 d. *INS* mRNA was measured by qPCR. (**C**) PANC-1 cells stably expressing RAC1^N17^ were subjected to TDC with IFNγ, IL1β, and TNFα (IIT) for 3 d and assayed for expression of *NEUROG3* by qPCR. (**D**) As in (**C**), except that PANC-1 cells were assayed for *INS*, *SLC2A2,* and *MAFA*. (**E**) PANC-1 cells were transiently transfected with RAC1^L61^, or empty pRK5 vector (V), subjected to TDC with FT (left-hand graph) or IIT (right-hand graph) and assayed for *INS* expression by qPCR. The data in (**A**–**E**) represent the mean ± SD of three assays each performed with technical replicates in triplicate. The asterisks (*) denote significant differences (*p* < 0.05, Wilcoxon test).

**Figure 2 cancers-13-05541-f002:**
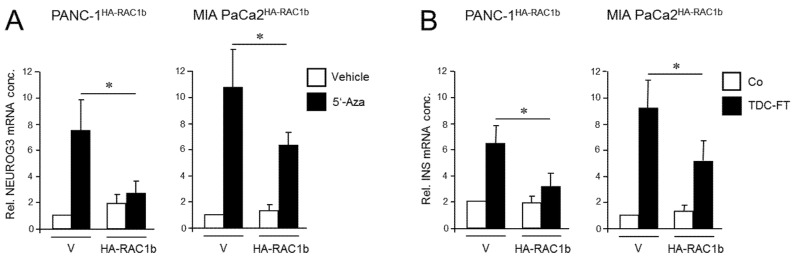
Ectopically expressed RAC1b blocks the conversion of PANC-1 and MIA PaCa-2 cells to cells with a pancreatic endocrine or β cell-like phenotype. (**A**) PANC-1 or MIA PaCa-2 cells with stable or transient expression, respectively, of HA-RAC1b were treated for 3 d with 5′-Aza (1 µM) in standard growth medium followed by another 3 d in growth medium without 5′-Aza. Cells were then subjected to qPCR for *NEUROG3*. Data are from three independent experiments and represent the mean ± SD. Asterisks (*) indicate a significant difference (*p* < 0.05, Wilcoxon test) relative to 5′-Aza-treated empty vector (V) transfected control cells. (**B**) As in (**A**) except that cells were subjected to FT-based TDC towards a β cell phenotype and assayed for *INS* expression. Control cells received standard growth medium. Data are from three independent experiments and represent the mean ± SD. Asterisks indicate a significant difference (*p* < 0.05, Wilcoxon test) relative to V transfected TDC-treated control cells.

**Figure 3 cancers-13-05541-f003:**
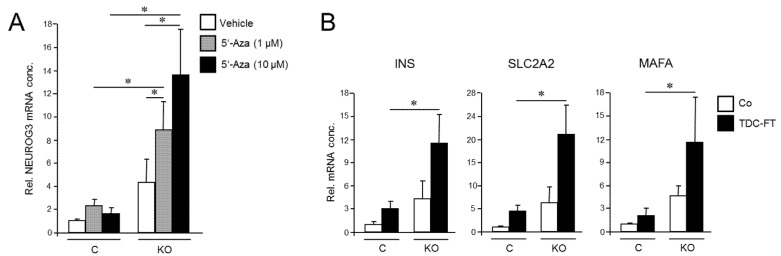
Cellular depletion of RAC1b enhances the conversion of PANC-1 cells into cells with a pancreatic endocrine or β cell-like phenotype. (**A**) PANC-1^RAC1b-KO^ (KO) and control (**C**) cells were treated or not for 3 d with two different concentrations of 5′-Aza in standard growth medium followed by another 3 d in growth medium without 5′-Aza. Cells were then subjected to qPCR for *NEUROG3*. Data are from three independent experiments and represent the mean ± SD. Asterisks (*) indicate significant differences (*p* < 0.05, Wilcoxon test). (**B**) As in (**A**), except that PANC-1^RAC1b-KO^ cells were subjected to FT-based TDC followed by qPCR for the indicated β cell markers. Data represent the mean ± SD of three (*INS*, *SLC2A2*) or four (*MAFA*) independent experiments. Asterisks (*) indicate significant differences (*p* < 0.05, Wilcoxon test).

**Figure 4 cancers-13-05541-f004:**
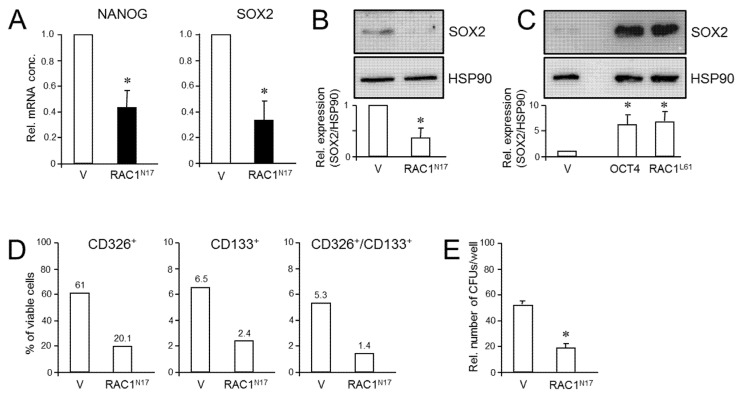
Effect of dn interference with RAC1 activation on stem cell marker expression and clonogenicity in PANC-1 cells. (**A**) PANC-1^RAC1-N17^ cells and empty vector controls (V) were assayed by qPCR for NANOG and SOX2 mRNA abundance. Data are the mean ± SD of three experiments. (**B**) As in (**A**) except that cells were analyzed by immunoblotting for SOX2. The graph underneath the blots depicts results from densitometry-based quantification of band intensities (mean ± SD, *n* = 3). (**C**) SOX2 immunoblotting of PANC-1 cells transfected with an expression for RAC1^L61^. As positive control, cells were transfected with an expression vector for OCT4, which is known to transcriptionally induce SOX2 expression [41]. (**D**) PANC-1^RAC1-N17^ cells were labeled singly or in combination with antibodies to CD326 or CD133, and measured by flow cytometry. Data shown are representative of three experiments. (**E**) PANC-1^RAC1-N17^ and V cells were seeded at a density of 40 cells/cm^2^ and incubated in standard growth medium for 10 d. Colonies formed were fixed with paraformaldehyde, stained with crystal violet, and CFUs >50 cells counted manually. Data shown are representative of three experiments (mean ± SD of triplicate wells). Asterisks (*) indicate a significant difference relative to controls (*p* < 0.05, Student’s *t*-test).

**Figure 5 cancers-13-05541-f005:**
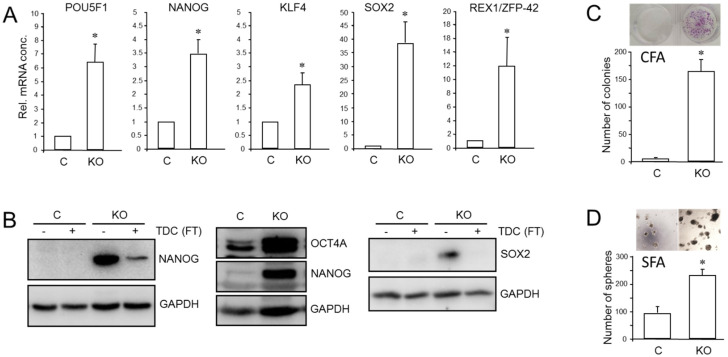
Effect of cellular depletion of RAC1b protein on stem cell marker expression and CSC generation in PANC-1 cells. (**A**) Expression of various stem cell-associated and self-renewal genes in PANC-1^RAC1b-KO^ and control cells as determined by qPCR. (**B**) Immunoblot analysis of NANOG, OCT4A, and SOX2 in PANC-1^RAC1b-KO^ cells. The blots shown are representative of three experiments. The left-hand and right-hand blots also contain lysates from cells that underwent TDC with FT for 5 d. (**C**) PANC-1^RAC1b-KO^ and control cells were seeded at a density of 40 cells/cm^2^ and incubated in normal growth medium for 10 d. Cells were then fixed, stained, and colonies >50 cells counted manually. Data are from two independent experiments and represent the mean ± SD of triplicate wells. (**D**) As in (**C**) except that 1000 cells/48-well were seeded in soft agar, allowed to grow for 14 d, and then counted manually under a microscope. Data are representative of two assays and represent the mean ± SD of five parallel wells. Asterisks (*) in (**A**,**C**,**D**) indicate a significant difference relative to control cells (*p* < 0.05, Student’s *t*-test).

**Figure 6 cancers-13-05541-f006:**
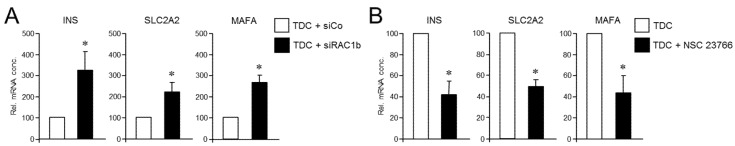
Reciprocal rescue experiments in PANC-1^RAC1-N17^ and PANC-1^RAC1b-KO^ cells and their effect on the TDC-induced β cell transcriptional phenotype. (**A**) PANC-1^RAC1-N17^ cells were transfected with 50 nM of either control (siCo) or RAC1b (siRAC1b) siRNA, subjected to FT-based TDC, and subsequently monitored for transcriptional activity of the INSULIN, GLUT2, and MaFA genes by qPCR. (**B**) As in (A), except that PANC-1^RAC1b-KO^ cells rather than being transfected were treated, or not, during TDC with 50 µM of the RAC1 inhibitor NSC 23766. Data in (**A**,**B**) are the mean ± SD (*n* = 3). The asterisks (*) denote significant differences relative to TDC + siCo or TDC controls (*p* < 0.05, Wilcoxon test).

**Figure 7 cancers-13-05541-f007:**
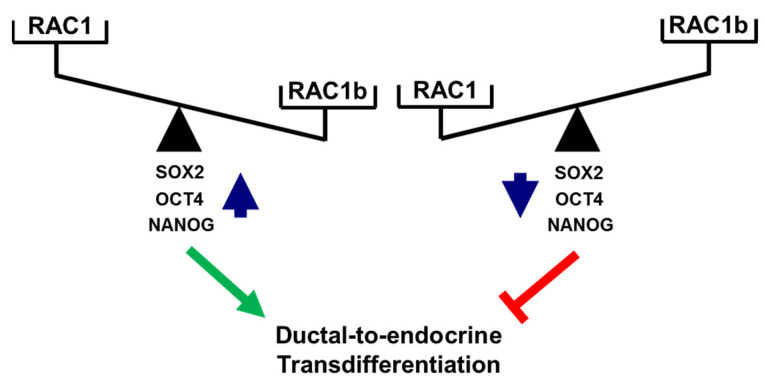
Cartoon illustrating antagonistic regulation of stemness factors and deTD potential in PANC-1 cells. High levels of activated RAC1 (or a high RAC1:RAC1b ratio) promote expression of SOX2, OCT4, and NANOG and favor a deTDtP, while the reverse is true for low levels of activated RAC1 (or a low RAC1:RAC1b ratio). The green arrow and red lines denote stimulatory and inhibitory effects, respectively.

## Data Availability

Data are contained within the article or Appendix A.

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
