# Peer review of "Activation of a Ductal-to-Endocrine Transdifferentiation Transcriptional Program in the Pancreatic Cancer Cell Line PANC-1 Is Controlled by RAC1 and RAC1b through Antagonistic Regulation of Stemness Factors"

_cancers, 2021, doi:10.3390/cancers13215541_

Round 1
Reviewer 1 Report
Cancers; title: Ductal-to-endocrine transdifferentiation of the pancreatic ductal adenocarcinoma cell line PANC-1 is controlled by RAC1 and RAC1b through antagonistic regulation of stemness factors.
Metastatic pancreatic ductal adenocarcinoma (PDAC) is very aggressive and poor prognosis. PDAC patients might have a need for replacement of insulin-producing cells, and conversion of tumor cells with a ductal/exocrine origin to endocrine cell-like cells. Schmidtlein P.M. et al. investigated the importance of RAC1 in the transdifferentiation of PNAC-1 and MIA PaCa-2 cell lines and clarified the impact of RAC1 and RAC1b on stem/pluripotency genes to control cellular plasticity. The purpose of this experiments is interesting, however, the results described here are not conclusive. Thus, this study is too preliminary, and the authors need a lot of experiments to reach the conclusion. The following experiments are required to publish.
[Main critiques]
- The author measured the deTD-transdifferentiation activity as the expressions of the indicated target mRNA and proteins only. This is not sufficient to conclude the transdiferentiation. The actual quantitation of the transdifferentiated cells was not done, and this differentiation is stable or transient (5 days?) or the terminus? These differentiated cells are fully functional or not?
For example, in 2.1 section, the transdifferentiation of PANC-1 and MIA PaCa-2 cells affected by RAC1 expression were represented by expressions of INS, SLC2A2, and MAFA mRNA levels; these mRNA cannot represent the actual protein expressions. Furthermore, no transdifferentiation results of cell staining, protein expression, cell functions, or morphology exhibited in this article. For this instance, authors need to advance the evidence in this article, or re-write more practical conclusions.
- In Figure 5, RAC1b KO showed the expression of stem cells markers mRNA and proteins like Oct4, Nanog, Klf4 and Sox2. However, this conversion to stemness should be done by generating the chimeric mice and analysis of the chimeric embryos.
- In Figure 7, the author insisted their hypothesis to balance the expression of RAC1 and RAC1b in stemness and transdifferentiation. However, the actual percentages of cells expressed the actual stemness and differentiation markers were not described at all and relative expression levels of RAC1 and RAC1b in these cells were not quantitated as well. It is hard to reach their conclusion.
- Seiz et al. reported recently that RAC1 and RAC1b used the different signaling (Biol Chem 2020, 401, 517-531), and epithelial splicing regulatory protein 1 (ESRP1) played a critical role of generating this differential transcript. These differential-signaling pathways and expression levels of ESRP1, should be examined and discussed.
- RAC1 and RAC1b were shown to affect the pluripotency marker expressed in PANC-1 cell line in the article. However, the mechanism and its potential pathway was not described in the article. Although knock down experiments and knock out experiments shown the expression of RAC1 and RAC1b are negative correlated with pluripotent markers, but it is insufficient to conclude that these effects were directly caused by RAC1, or RAC1b repression.
- To prove the importance of RAC1 and RAC1b in transdifferentiation, rescue experiments of knock down of RAC1, RAC1b are recommended to carry out to obtain the conclusive one.
[Minor critiques]
- There are no staining data of indicated cells with stemness and differentiation specific markers such as Oct4, Sox2, Klf4, Nanog, Ins, Slc2A2, Mafa ets.
- In Figure 4B and C, the author used anti-HSP90, Santa Cruz antibodies. It includes HSP90alpha and beta, indicating the data od Western blotting using this antibody was not conclusive. This showed two mixed proteins as the controls which are not good enough as a control. Please use the different control antibodies. In addition, RACL61 seemed to increase the expression of HSP90.
- Regarding the dominant negative (N17) and constitutive active (L61) RAC1 expression data, the authors did not show any expression levels of each construct in PANC1 cells. In addition, the forced expression of these constructs does not affect the endogenous expression of RAC1 and RAC1b ?
- In Figure 1, the effects of TDC-FC and TDC-IIT in RANC-1 or MIAPaCa-2 cells were examine. However, they showed only the expressions of mRNA of INS, SLCA2, MAFA and NEUROG3. Please exhibit the staining data of TD using the differentiation marker proteins.
- In Figures 2 and 3, the treatment by 5-Aza and TDC-FT were described. However, the methylation status of the NEUROG3 (NGN3?) genome should be analyzed to confirm the conclusion.
- In Figure 6, the author used NSC23766, RAC1 inhibitor to block the TDC-induced beta cell transcriptions. The data of RAC1 binding and activation by the RAC-specific GEF Trio or Tiam1 should be generated in this RAC1b-deficient PANC-1 cells to demonstrate the introduced genes are really functional.
- The protein expression levels of HA-RAC1b introduced PANC-1 and MIA PaCa2 cells should be demonstrated in the presence or absence of 5’-Aza or FT treatment.
Author Response
Reviewer1
Metastatic pancreatic ductal adenocarcinoma (PDAC) is very aggressive and poor prognosis. PDAC patients might have a need for replacement of insulin-producing cells, and conversion of tumor cells with a ductal/exocrine origin to endocrine cell-like cells. Schmidtlein P.M. et al. investigated the importance of RAC1 in the transdifferentiation of PNAC-1 and MIA PaCa-2 cell lines and clarified the impact of RAC1 and RAC1b on stem/pluripotency genes to control cellular plasticity. The purpose of this experiments is interesting, however, the results described here are not conclusive. Thus, this study is too preliminary, and the authors need a lot of experiments to reach the conclusion. The following experiments are required to publish.
[Main critiques]
- The author measured the deTD-transdifferentiation activity as the expressions of the indicated target mRNA and proteins only. This is not sufficient to conclude the transdiferentiation. The actual quantitation of the transdifferentiated cells was not done, and this differentiation is stable or transient (5 days?) or the terminus? These differentiated cells are fully functional or not?
For example, in 2.1 section, the transdifferentiation of PANC-1 and MIA PaCa-2 cells affected by RAC1 expression were represented by expressions of INS, SLC2A2, and MAFA mRNA levels; these mRNA cannot represent the actual protein expressions. Furthermore, no transdifferentiation results of cell staining, protein expression, cell functions, or morphology exhibited in this article. For this instance, authors need to advance the evidence in this article, or re-write more practical conclusions.
Response: We perfectly agree with the reviewer that the expression of endocrine and β cell markers was mainly studied at the mRNA level. Our focus was on the demonstration that our transdifferentiation conditions are able to activate a transcriptional program of endocrine and β cell development rather than to induce the expression of the respective proteins. To emphasize this, we have now defined the term “transdifferentiation” more carefully by introducing the term “ductal-to-endocrine transdifferentiation transcriptional program” (deTDtP). It should be noted here that in almost all studies that attempted to generate b cell-like cells, markers were detected only at the mRNA level, except for INS, probably because expression was generally low.
Quantification of transdifferentiation at the mRNA level was done at the cell population but not at the single-cell level. As outline above, detection of proteins is difficult due to their low expression. In the case of insulin, we have attempted to measure it by ELISA in conditioned supernatants from transdifferentiated PANC-1 cells, but like other investigators (PMID: 29700299, see below) could not reliably detect it. The morphology of ductal-to-endocrine transdifferentiated cells PANC-1 and MIA PaCa-2 cells has been shown by others (Ref. 13) and by us in our previous paper (Ref. 23). It should also be pointed out that other studies relied on the measurement of only a single gene or protein to demonstrate successful TD, either INS, or NEUROG3 (Refs. 6+13), while in our study a panel of four different marker genes was quantified.
Since protein quantification at the single cell level, i.e., by immunocytochemistry, is not feasible, we sought another strategy to quantify transdifferentiation at the single cell level. We generated single clones from PANC-1 cultures by limited dilution and after transdifferentiation culture with IIT analysed them separately by qPCR for INS expression. Of note, all clones induced the INS gene albeit to a different extent revealing heterogeneity in plasticity/transdifferentiation potential within the PANC-1 cell population. These data have been included in the Supplementary material as Figure S1D and briefly discussed in paragraph 5 of section 4.
Regarding the stability of the transdifferentiated phenotype, we designed an experiment where we transdifferentiated PANC-1 cells with IFNγ, IL1β and TNFa (IIT) for 3 days and instead of terminating the experiment by cell lysis replaced the differentiation medium by normal growth medium (without IIT) and extended culture for up to 7 days with samples taken every 2-3 days. The results show that INS mRNA levels remained elevated for the entire period, suggesting that the β cell-like phenotype is stable. These data were added to the Supplementary material as Figure S1C.
With respect to functionality, we believe that the differentiated cells are likely not fully functional given the well-known challenges in generating from stem cells by directed differentiation genuine functional β-like cells in vitro (see PMID: 33815669, PMID: 32587391). It has been found that the pancreatic β cell-like cells obtained by in vitro differentiation still have many defects, including lack of adult-type glucose stimulated insulin secretion, and multi-hormonal secretion, suggesting that in vitro culture does not allow for obtaining fully mature β-like cells for transplantation. With respect to our cell system, it remains to be determined to what extent functional activities typical of mature b cells are exhibited by transdifferentiated PANC-1 or MIA PaCa-2 cells. However, we feel that these experiments are beyond the scope of this study.
- In Figure 5, RAC1b KO showed the expression of stem cells markers mRNA and proteins like Oct4, Nanog, Klf4 and Sox2. However, this conversion to stemness should be done by generating the chimeric mice and analysis of the chimeric embryos.
Response: We appreciate the reviewers’ suggestion, however, the generation of chimeric mice is expensive and time-consuming. The increase in the expression of stem cell markers in PANC-1-RAC1b-KO cells was associated with functional stemness features that were clearly demonstrated here by the large increase in colony/sphere numbers in well-established stem cell assays like CFA and SFA (see Figure 5C+D).
- In Figure 7, the author insisted their hypothesis to balance the expression of RAC1 and RAC1b in stemness and transdifferentiation. However, the actual percentages of cells expressed the actual stemness and differentiation markers were not described at all and relative expression levels of RAC1 and RAC1b in these cells were not quantitated as well. It is hard to reach their conclusion.
Response: Our hypothesis on regulation of stemness and plasticity through balanced expression of RAC1 and RAC1b was based on functional data at the cell population level. We have performed relative comparisons of expression at the population level. Rather than performing immunocytochemistry to determine the percentage of cells expressing differentiation markers, we have shown above that 6/6 single cell-derived clones of (non-genetically modified) PANC-1 cell cultures contributed to elevated insulin expression albeit to a varying extent (see above, new Figure S1D). We employed the same strategy here with PANC-1-RAC1b-KO cells in order to describe the actual percentages of cells that express the actual stemness markers. Single clone analysis revealed that all clones of PANC-1RAC1b-KO had increased SOX2 and OCT4A expression, again with some variations among clones (see new Figure S3A).
Data on the expression levels of RAC1 and RAC1b and a quantification of their non-activated and activated fractions in PANC-1 cells under standard culture conditions have been published by us previously (PMID: 24378395, Ref. 37).
PANC-1RAC1b-KO cells are devoid of RAC1b protein and have unaltered RAC1 protein levels (refer to our previous paper, PMID: 31108998, Ref. 38). We, therefore, believe that based on the data in the revised manuscript version our conclusions/hypothesis summarised in Figure 7 are valid.
- Seiz et al. reported recently that RAC1 and RAC1b used the different signaling (Biol Chem 2020, 401, 517-531), and epithelial splicing regulatory protein 1 (ESRP1) played a critical role of generating this differential transcript. These differential-signaling pathways and expression levels of ESRP1, should be examined and discussed.
Response: We agree with the reviewer that this paper nicely studied the different signaling pathways activated by the two RAC1 isoforms and it is definetely important to analyse which of these are involved in mediating the effects of RAC1 and RAC1b on deTD. However, we feel that their in-depth analysis is beyond the scope of this study. Likewise, the issue of how the generation/splicing of RAC1 and RAC1b transcripts is controlled by expression of ESRP1 is surely interesting but would not provide additional information as to how RAC1 and RAC1b antagonistically regulate stemness genes and deTD. Nevertheless, based on the report by Seiz et al. we have briefly discussed potential signaling pathways involved (fourth paragraph of section 4) as most of them are also involved in regulating stemness (for review see PMID: 27611937).
In this context, we should mention our recent observation that differential regulation of RAC1/RAC1b generation/splicing is controlled by TGFb, probably through regulation of ESRP1 expression. These data are supposed to be included in a separate manuscript.
- RAC1 and RAC1b were shown to affect the pluripotency marker expressed in PANC-1 cell line in the article. However, the mechanism and its potential pathway was not described in the article. Although knock down experiments and knock out experiments shown the expression of RAC1 and RAC1b are negative correlated with pluripotent markers, but it is insufficient to conclude that these effects were directly caused by RAC1, or RAC1b repression.
Response: We very much appreciate the comment of the reviewer that elucidating the (signaling) pathways triggered by RAC1/RAC1b is an important task, however, an identification these pathways is beyond the scope of this study (as outlined under 4.). We did not strive to identify the signaling pathways triggered by of RAC1 and RAC1b. We agree with the reviewer that the positive effects of RAC1 (and the negative effects of RAC1b) on the pluripotency markers need not be direct. At one location in our manuscript where we stated that effects are direct (third paragraph in the Discussion), we have removed this statement. For discussion of potential signaling pathways see above (point 4).
- To prove the importance of RAC1 and RAC1b in transdifferentiation, rescue experiments of knock down of RAC1, RAC1b are recommended to carry out to obtain the conclusive one.
Response: We perfectly agree with the reviewer that the suggested rescue approaches via knockdown of RAC1/RAC1b would strengthen our conclusions. Figure 6 shows results of such a (non-genetic) rescue experiment. Here, the inductive effect of RAC1b-KO on deTDtP was suppressed by application of NSC 23766. Unfortunetely, inhibition of RAC1 by a RAC1-selective small interfering RNA is not feasible given that all nucleic acid sequences of RAC1 are also present in RAC1b. In addition, we have now performed the reverse rescue experiment by transfecting PANC-1RAC1-N17 cells with a RAC1b/exon 3b-selective siRNA. Intriguingly, this re-induced the expression of INS, SLC2A2 and MAFA and hence partially prevented their RAC1N17-mediated downregulation. These new data have been added to Figure 6 as panel A, while the former Figure 6 became panel B. The corresponding part in the Discussion section (fourth paragraph) has been modified accordingly.
[Minor critiques]
- There are no staining data of indicated cells with stemness and differentiation specific markers such as Oct4, Sox2, Klf4, Nanog, Ins, Slc2A2, Mafa ets.
Response: As mentioned above, we have analysed individual clones of transdifferentiated PANC-1 cells by qPCR for INS expression, and PANC-1RAC1b-KO cells by immunoblotting for SOX2 and OCT4 expression and found that all clones exhibited elevated levels of INS mRNA (relative to untreated controls) or SOX2 protein (relative to vector controls), albeit with some variation among clones. These data are part of Supplementary material of the revised version and shown in Figures S1B and S4A.
- In Figure 4B and C, the author used anti-HSP90, Santa Cruz antibodies. It includes HSP90alpha and beta, indicating the data od Western blotting using this antibody was not conclusive. This showed two mixed proteins as the controls which are not good enough as a control. Please use the different control antibodies. In addition, RACL61 seemed to increase the expression of HSP90.
Response: We should point out here that we have used this anti-HSP90 antibody in numerous studies in Western blot analyses of PANC-1 cells. It is a well-established antibody that generates a single band and was used in 524 publications (see https://www.scbt.com/de/p/hsp-90alpha-beta-antibody-f-8). We would, therefore, prefer to leave the blots on Figures 4B and C as they stand. In various other blots, we have used either anti-GAPDH or anti-b-actin antibodies. The slightly higher abundance of HSP90 in the RAC1-L61 lane merely reflects higher total protein loading rather than a specific effect of RAC1-L61 on HSP90 protein levels.
- Regarding the dominant negative (N17) and constitutive active (L61) RAC1 expression data, the authors did not show any expression levels of each construct in PANC1 cells. In addition, the forced expression of these constructs does not affect the endogenous expression of RAC1 and RAC1b ?
Response: The reviewer is correct in that we did not provide expression levels of both mutants. Stable overexpression of RAC1N17 in the PANC-1 cells used in this study has been shown in a previous publication (Ref. 36) as mentioned in the Results (section 2.1) and in M&M (section 4.1). However, we have included two blots showing results from anti-MYC immunoprecipitation experiments of MYC-tagged versions of RAC1N17 after transfection into PANC-1 or MIA PaCa-2 cells (Figure S1A, left-hand panel and middle panel), and of MYC-RAC1L61 following transfection into PANC-1 cells (Figure S1A, right-hand panel).
In addition, forced expression of either RAC1N17 or RAC1L61 did not alter levels of endogenous RAC1 (shown for RAC1L61 in the new right-hand panel of Figure S1A) or RAC1b. The migration data have been renumbered as Figure S1B.
- In Figure 1, the effects of TDC-FC and TDC-IIT in RANC-1 or MIAPaCa-2 cells were examine. However, they showed only the expressions of mRNA of INS, SLCA2, MAFA and NEUROG3. Please exhibit the staining data of TD using the differentiation marker proteins.
Response: Please see my responses to main critiques, point 1 and minor critiques, point 1. Since protein expression levels are generally too low to be safely quantified, we have focused our study on the transcriptional level.
- In Figures 2 and 3, the treatment by 5-Aza and TDC-FT were described. However, the methylation status of the NEUROG3 (NGN3?) genome should be analyzed to confirm the conclusion.
Response: This is quite an interesting suggestion. However, 5’-Aza is a well established demethylating agent for epigenetic reprogramming and we have used it according to a published protocol specifically established for PANC-1 cells (PMID: 19913512, Ref. 13). However, another study (PMID: 29700299) with PANC-1 cells found that the INS, GCG, and IAPP genes were significantly increased in 5’-Aza-treated PANC-1 cells. Interestingly, these authors were able to show that the reexpression of both INS (and SST, encoding the somatostatin gene) was the consequence of demethylation of the CpG sites located in the promoters of these genes. Given this demonstration, which we have briefly referenced (#67, last paragraph of the Discussion), we believe that carrying out methylation assays in our study is not mandatory.
In this context, it is interesting to note that in the same study (PMID: 29700299), the authors attempted to detect insulin protein in 5’-Aza-reprogrammed PANC-1 cells using the ELISA method, however, they were not successful in this respect. We should mention here that we also failed to detect insulin by ELISA in culture supernatants from transdifferentiated PANC-1 cells. The authors speculate that the SST augmentation observed in the PANC-1 cells might participate in the alteration of INS production at the protein level because of its inhibitory effect. They believe that their data support the importance of further developing epigenetic methods to induce the differentiation of PDAC cell lines toward an endocrine-like lineage in order to generate relevant insulin-producing human cell models.
- In Figure 6, the author used NSC23766, RAC1 inhibitor to block the TDC-induced beta cell transcriptions. The data of RAC1 binding and activation by the RAC-specific GEF Trio or Tiam1 should be generated in this RAC1b-deficient PANC-1 cells to demonstrate the introduced genes are really functional.
Response: NSC 23766 is a well-established inhibitor. We therefore believe that carrying out RAC1 binding and activation assays is not required to further support our conclusions inasmuch as we are not interested here in the specific roles played by Trio or Tiam1. Most studies that employed this inhibitor did not show blocking of binding/activation of RAC1 by Trio or Tiam1 (PMIDs x, y, z). The functionality of this inhibitor in PANC-1 cells has been demonstrated by us previously (Ref. 36). Moreover, the validity of the RAC1 inhibitory effect was confirmed by the RAC1b-selective siRNA-mediated knockdown experiment (see main critiques, point 6). In this experiment, no genes have been introduced into RAC1b-deficient PANC-1 cells! The functional consequences of the RAC1b (= RAC1 exon 3b) knockout have been extensively characterized in previous publications from our group, i.e., Ref. 38.
- The protein expression levels of HA-RAC1b introduced PANC-1 and MIA PaCa2 cells should be demonstrated in the presence or absence of 5’-Aza or FT treatment.
Response: As requested, we have carried out immunoblots with PANC-1 and MIA PaCa-2 cells stably and transiently, respectively, expressing HA-RAC1b and treated, or not, with 5’-Aza or FT. These blots showed that neither 5’-Aza nor FT treatment had significant effects on the protein levels of the HA-RAC1b transgene. These data have been added to the Suppl. material as Figure S2.
Additional changes made:
- Please note change of title.
- In the course of removing duplications, we have condensed the text at some locations. This has been highlighted in green.
Reviewer 2 Report
Methods: The sequences for the primers used for qRT-PCR need to be provided.
There are so many abbreviations in the paper that it is hard to follow. Also TD (Transdifferentiation) is used in several contexts which makes it also confusing and perhaps these should be spelled out (TDC), TDC-FT, TDC- IIT
The specific genes and their names should be spelled out the first time they are used: i.e., SLC2A2, MAFA and what proteins they code for.
Can the authors comment on the relationship to RAC1 and the RhoA signaling pathway involved with gastrin and paxillin in EMT. The paper [Mu, G., Ding, Q., Li, H. et al. Gastrin stimulates pancreatic cancer cell directional migration by activating the Gα12/13–RhoA–ROCK signaling pathway. Exp Mol Med 50, 1–14 (2018)]. Also used PANC-1 cells and studied EMT.
Please clarify the difference between the RAC1-T17N described as a stable clone in this paper and the paper referenced (Ref 36) called “Rac1-N17”. Are they the same clone or different, pleas elaborate.
Line 145: “Similar results for INS expression were obtained with MIA PaCa-2 cells transiently transfected with a RAC1N17-encoding expression vector (Figure 1B)”. Since only INS was examined in MIA Paca2 cells, please change sentence to state that the INS expression was similar in MIA Paca2 and PANC-1 if there other genes were not compared. The Figures in Figure 1 should be labeled as to which cell line (PANC-1 or MIA PaCa2 ) similar to Figure 2.
On Fig 1B change the INS expression to under the x-axis like figure 1A for consistency.
Line 194: “PANC-1 cells stably expressing RAC1b (PANC-1HA- RAC1b) have been characterized previously and shown to suffer from reduced migratory activity [37].” Cells don’t suffer. Please reword. i.e., these cells have reduced migration.
Figure 2 legend: Asterix should denote the value of the p value. (state P<0.05)
Figure 3 is missing p value: “Asterisks () indicate significant differences.
Figure 4 legend “and incubated in normal growth medium” – what is normal versus abnormal?
Figure 4D: please show the flow images for CD133.
Comparisons should not be made to PaTu 8988s cells if these were not used in this investigation.
There is no description in the methods of details for developing PANC-1 cells that lack RAC1b protein by removing the RAC1b- 221specifying exon 3b from RAC1 using CRISPR/Cas9 technology.
Author Response
Reviewer 2
Comments and Suggestions for Authors
Methods: The sequences for the primers used for qRT-PCR need to be provided.
Response: All primer sequences have been published in previous publications from our group. However, we have supplied along with the revised manuscript a table (Table S1) with all primers used in the present study.
There are so many abbreviations in the paper that it is hard to follow. Also TD (Transdifferentiation) is used in several contexts which makes it also confusing and perhaps these should be spelled out (TDC), TDC-FT, TDC- IIT
Response: As requested, the term “transdifferentiation” when used alone is now spelled out. However, we would prefer to keep the abbreviations deTDC, TDC-FT and TDC-IIT as these appear many times in our manuscript. We have also introduced the abbreviation “deTDtP” (for ductal-to-endocrine transdifferentiation transcriptional program) in response to a request from
The specific genes and their names should be spelled out the first time they are used: i.e., SLC2A2, MAFA and what proteins they code for.
Response: As requested, full names were given together with the names of the respective proteins (second paragraph of the Introduction).
Can the authors comment on the relationship to RAC1 and the RhoA signaling pathway involved with gastrin and paxillin in EMT. The paper [Mu, G., Ding, Q., Li, H. et al. Gastrin stimulates pancreatic cancer cell directional migration by activating the Gα12/13–RhoA–ROCK signaling pathway. Exp Mol Med 50, 1–14 (2018)]. Also used PANC-1 cells and studied EMT.
Response:
Rac1 and RhoA are both indispensable for cell migration as they regulate cell protrusion, cell-extracellular matrix interactions and force transduction. the interactions between paxillin (and vinculin and Hic-5) are spatially and reciprocally regulated by the activity of Rac1 and RhoA. Here, paxillin interacts with active vinculin in adhesions in response to Rac1 and RhoA activation, while inactive vinculin interacts with paxillin in the membrane following Rac1 inhibition. Additionally, Rac1 specifically regulates the dynamics of paxillin in adhesions [PMID 22629471]. Paxillin also coordinates the spatiotemporal activation of signaling molecules, including Rac1 and RhoA GTPases by recruiting GEFs, GAPs, and GITs to focal adhesions [PMID 28214467]. Rac1 and RhoA can show opposite behaviors and spatial localisations, with RhoA being active toward the rear of the cell and regulating its retraction during migration, whereas Rac1 is active toward the front of the cell. Moreover, RhoA and Rac1 activities at the leading edge of the cells peak at separate points during the migratory cycle of protrusion and retraction [PIMD 27533896]. With respect to a connection among RhoA, RAC1 and EMT it was shown that interferon regulatory factor 4 binding protein (IBP), NKCC1, mTOR, mTORC1 and mTORC2 regulate EMT at least in part by downregulation of RhoA and Rac1 signaling pathways [PMID 23975422, PMID 30159893, PMID 25043657, PMID 21430067]. With regard to Rac and gastrin there was only one entry in PubMed reporting that (glycine-extended) gastrin stimulates mouse gastric epithelial IMGE-5 cell proliferation and migration through Rho/ROCK- but not Rac-dependent pathways [PMID 15845872].
Please clarify the difference between the RAC1-T17N described as a stable clone in this paper and the paper referenced (Ref 36) called “Rac1-N17”. Are they the same clone or different, pleas elaborate.
Response: Both terms refer to the same mutant. The “T” indicates the amino acid in the wild-type sequence of Rac1. However, we have removed the term RAC1-T17N to avoid this confusion.
Line 145: “Similar results for INS expression were obtained with MIA PaCa-2 cells transiently transfected with a RAC1N17-encoding expression vector (Figure 1B)”. Since only INS was examined in MIA Paca2 cells, please change sentence to state that the INS expression was similar in MIA Paca2 and PANC-1 if there other genes were not compared. The Figures in Figure 1 should be labeled as to which cell line (PANC-1 or MIA PaCa2 ) similar to Figure 2.
Response: As requested, the wording/statement in line 145 has been changed. In addition, the various panels in Fig. 1 were labeled with cell line names.
On Fig 1B change the INS expression to under the x-axis like figure 1A for consistency.
Response: As requested, the graphs in Figures 1B, C and E were labeled below the x-axis to conform with the style in panel A.
Line 194: “PANC-1 cells stably expressing RAC1b (PANC-1HA- RAC1b) have been characterized previously and shown to suffer from reduced migratory activity [37].” Cells don’t suffer. Please reword. i.e., these cells have reduced migration.
Response: As requested, the wording has been changed.
Figure 2 legend: Asterix should denote the value of the p value. (state P<0.05)
Response: As requested, the value (p<0.05) has been added.
Figure 3 is missing p value: “Asterisks (b) indicate significant differences.
Response: This has been corrected.
Figure 4 legend “and incubated in normal growth medium” – what is normal versus abnormal?
Response: “Normal” is supposed to mean standard growth medium, the composition of which is given in section 4.1. The term “normal” has been replaced by “standard” throughout the manuscript.
Figure 4D: please show the flow images for CD133.
Response: Due to the low number of positive cells (6.5% in PANC-1 vector controls vs 2.4% in PANC-1-RAC1-N17 cells), an illustration by showing the flow images is not feasible. However, we have included - for conviction of the reviewer only - a histogram plot and the gating strategy for detection of CD133 in the vector control cells.
Comparisons should not be made to PaTu 8988s cells if these were not used in this investigation.
Response: This comparison relates to data of this cell line in a previous paper (Ref. 23). However, we have deleted this cell line from the comparison.
There is no description in the methods of details for developing PANC-1 cells that lack RAC1b protein by removing the RAC1b- 221specifying exon 3b from RAC1 using CRISPR/Cas9 technology.
Response: This is correct, however, the description and full characterization is given in a previous publication, which we referred to (Ref. 38).
Additional changes made:
- Please note change of title.
- In the course of removing duplications we have condensed the text at some locations. This has been highlighted in green.
Reviewer 3 Report
Hendrik Ungefroren et al. uncovered RAC1 and RAC1b to antagonistically control growth factor-induced activation of an endocrine transcriptional program and the generation of CSCs in PDAC cells.
Point to be discussed:
1. for all Western blot figures, densitometry readings/intensity ratio of each band should be included; the whole Western blot showing all bands and molecular weight markers should be included in the Supplementary Materials
Fig. 5 c,d: did the authors normalized for cell number, viability and proliferation index?
Fig 7: can be optimized from a graphical standpoint, making closer to a graphical abstract with cell behaviour schematic representation.
This author personally misses some important implication for a broader interest in the oncology field:
Low oxygen concentration and vascular system are key in order to keep the homeostasis of CSCs. Since anti-vascular endothelial growth factor receptor (anti-VEGFR) pioneered the attempts to normalize tumor vasculature and restore its function, as indicated by tissue perfusion and decreasing intratumoral hypoxia, further investigations aimed at matching the authors report with shaping the intratumoral immune cell phenotype in parallel with vascular normalization can be relevant. Indeed, vascular endothelial growth factor (VEGF) and inflammatory molecules are not merely key proangiogenic elements, but are also immune modulators, boost vascular formation and cooperate in creating permissive environment in most lethal malignancies, and lead to poor drug response and impact survival, potentially regulating EMT and stemness factors (can the authors expand introduction and discussion section referring to PMID: 33203154?)
Author Response
Hendrik Ungefroren et al. uncovered RAC1 and RAC1b to antagonistically control growth factor-induced activation of an endocrine transcriptional program and the generation of CSCs in PDAC cells.
Point to be discussed:
- for all Western blot figures, densitometry readings/intensity ratio of each band should be included; the whole Western blot showing all bands and molecular weight markers should be included in the Supplementary Materials
Response: We had submitted this file along with the manuscript, however, the journal may not have passed it to this reviewer. Accompanying the revised version is another copy of this file, which now also contains the densitometry readings/intensity ratio of bands from those Western blots for which quantitative data are shown in the manuscript.
Fig. 5 c,d: did the authors normalized for cell number, viability and proliferation index?
Response: This is a good question. We took extreme care to be sure that identical numbers of viable cells were seeded (briefly mentioned and emphasized now in section 4.5 of the revised version), since normalization for cell numbers at the end of the assay is not possible. Since we counted only clones/spheres that were adherent (CFA) or contained sufficient numbers of cells to be visible under the microscope (CFA and SFA), these cells were all viable. The proliferation index of the stem cell-like cells that gave rise to colonies may differ slightly as inferred from variations in colony size at the time of assay termination.
Fig 7: can be optimized from a graphical standpoint, making closer to a graphical abstract with cell behaviour schematic representation.
Response: We prefer to leave this cartoon as is for the reasons of clarity and simplicity.
This author personally misses some important implication for a broader interest in the oncology field: Low oxygen concentration and vascular system are key in order to keep the homeostasis of CSCs. Since anti-vascular endothelial growth factor receptor (anti-VEGFR) pioneered the attempts to normalize tumor vasculature and restore its function, as indicated by tissue perfusion and decreasing intratumoral hypoxia, further investigations aimed at matching the authors report with shaping the intratumoral immune cell phenotype in parallel with vascular normalization can be relevant. Indeed, vascular endothelial growth factor (VEGF) and inflammatory molecules are not merely key proangiogenic elements, but are also immune modulators, boost vascular formation and cooperate in creating permissive environment in most lethal malignancies, and lead to poor drug response and impact survival, potentially regulating EMT and stemness factors (can the authors expand introduction and discussion section referring to PMID: 33203154?)
Response: We agree with this reviewer regarding the importance of these processes. As requested, we have briefly discussed these issues in regard to the connection of CSCs with hypoxia and VEGF in at the end of paragraph 5 in the Discussion section based on the suggested publication.
Additional changes made:
- Please note change of title.
- In the course of removing duplications we have condensed the text at some locations. This has been highlighted in green.
Round 2
Reviewer 1 Report
Now the authors responded all questions. This MS should be accepted.
Reviewer 2 Report
The authors have appropriately addressed all the concerns of this reviewer. The manuscript reads much more smoothly.
Reviewer 3 Report
The authors have clarified several of the questions I raised in my previous review. Most of the major problems have been addressed by this revision.